# Disentangling conical intersection and coherent molecular dynamics in methyl bromide with attosecond transient absorption spectroscopy

Henry Timmers[1,9], Xiaolei Zhu [2,3,9], Zheng Li [2,3,4,5,9], Yuki Kobayashi [1], Mazyar Sabbar[1], Maximilian Hollstein[6], Maurizio Reduzzi[1], Todd J. Martínez [2,3], Daniel M. Neumark [1,7] & Stephen R. Leone[1,7,8]

Attosecond probing of core-level electronic transitions provides a sensitive tool for studying valence molecular dynamics with atomic, state, and charge specificity. In this report, we employ attosecond transient absorption spectroscopy to follow the valence dynamics of strong-field initiated processes in methyl bromide. By probing the $3d$ core-to-valence transition, we resolve the strong field excitation and ensuing fragmentation of the neutral $\sigma^\star$ excited states of methyl bromide. The results provide a clear signature of the non-adiabatic passage of the excited state wavepacket through a conical intersection. We additionally observe competing, strong field initiated processes arising in both the ground state and ionized molecule corresponding to vibrational and spin-orbit motion, respectively. The demonstrated ability to resolve simultaneous dynamics with few-femtosecond resolution presents a clear path forward in the implementation of attosecond XUV spectroscopy as a general tool for probing competing and complex molecular phenomena with unmatched temporal resolution.

[1] Department of Chemistry, University of California, Berkeley, CA 94720, USA. [2] Department of Chemistry and The PULSE Institute, Stanford University, Stanford, CA 94305, USA. [3] SLAC Linear Accelerator Laboratory, Menlo Park, CA 94025, USA. [4] Max Planck Institute for the Structure and Dynamics of Matter, 22761 Hamburg, Germany. [5] Department of Physics, Peking University, 100871 Beijing, China. [6] Department of Physics, University of Hamburg, 20355 Hamburg, Germany. [7] Chemical Sciences Division, Lawrence Berkeley National Laboratory, Berkeley, CA 94720, USA. [8] Department of Physics, University of California, Berkeley, CA 94720, USA. [9]These authors contributed equally: Henry Timmers, Xiaolei Zhu, Zheng Li. Correspondence and requests for materials should be addressed to S.R.L. (email: srl@berkeley.edu)

Photo-initiated dynamics in the valence shells of molecules form the basis for understanding molecular photophysics. These valence dynamics typically follow a geometric reaction coordinate in an excited molecular state. The timescale for such interatomic, or nuclear, motion occurs within the multi-femtosecond time domain (>1 fs)[1]. As a result, femtosecond spectroscopy has become a well-established technique to probe a range of fundamental molecular dynamics, including light harvesting in photosynthetic compounds[2,3] and photocatalytic charge transfer reactions[4]. Femtosecond spectroscopy has also been applied to study the regime where the nuclear dynamics become coupled with the electronic degrees of freedom within a molecule, leading to non-adiabatic transitions between different electronic states mediated by multi-dimensional conical intersections[5–9]. However, excited-state molecular dynamics can often be quite intricate, involving numerous non-adiabatic transitions. This complexity cannot always be captured with coarse femtosecond time resolution. Therefore, to resolve unambiguously the complete evolution of molecular excited state wavepackets, even finer temporal resolution is required.

Isolated attosecond, extreme ultraviolet (XUV) pulses generated through the process of high harmonic generation (HHG)[10] provide the perfect tool for investigating such short time photochemical dynamics. The use of isolated attosecond pulses over the past decade has helped to resolve non-adiabatic dynamics in both ionic[11,12] and highly-excited neutral[13] molecules with unprecedented time resolution. In addition, the temporal resolution provided by these pulses has enabled the study of electron motion in atomic[14,15], molecular[16,17], and solid-state[18,19] systems.

In this report, we use attosecond transient absorption spectroscopy (ATAS) to probe the neutral excited state dynamics of methyl bromide ($CH_3Br$) with few-femtosecond temporal resolution. Numerous previous studies have investigated the velocity distributions of photofragments generated upon ultraviolet excitation and dissociation from the $\sigma^\star$ excited A-band of $CH_3Br$[20–22]. These energy-resolved studies have found that a conical intersection between the $^3Q_{0+}$ and $^1Q_1$ states gives rise to a non-adiabatic transition probability of ~15–30% from the $^3Q_{0+}$. However, no study has directly probed these non-adiabatic dynamics in real time for $CH_3Br$.

To resolve these neutral excited-state dynamics with ATAS, a few-femtosecond near infrared (NIR) pump pulse is used to initiate a valence excited-state wavepacket in the molecule, in this case via strong field excitation. The resulting dynamics are probed using a time-delayed attosecond, XUV pulse resonant with the Br $M_{4,5}$ core-to-valence transitions, leading to the absorption of XUV photons. These core level absorption edges exhibit element, charge, and state specificity and can therefore disentangle intricate, non-adiabatic dynamics. Utilizing this scheme, we resolve the complete evolution of the excited-state wavepacket, from the initial excitation to the ensuing fragmentation. Due to the exquisite time resolution provided by ATAS, the results provide a signature of a non-adiabatic passage through a conical intersection, verified by molecular wavepacket propagation simulations. In addition, we simultaneously probe competing dynamics initiated by the strong field excitation pulse, including the creation and evolution of a ground-state vibrational wavepacket and the coherent motion of a spin-orbit wavepacket in the ionized parent molecule. The ability to simultaneously resolve competing molecular dynamics with attosecond time resolution demonstrates the potential of this unique spectroscopic technique.

## Results

**Probing strong field processes near the Br $M_{4,5}$-edge.** To initiate and observe the strong-field valence processes in $CH_3Br$, we use the pump-probe scheme depicted in Fig. 1. A 4 fs, carrier-envelope phase stable NIR field is focused into a gas cell with a 2 mm path length and filled with 10 torr of $CH_3Br$. The valence molecular orbital configuration of $CH_3Br$ is given by $(a_1)^2(e)^4$. The intensity reached by the NIR field, $I = 1.5 \times 10^{14}$ W/cm², is sufficient to distort the molecular potential, leading to strong field ionization of the valence (e) electron. While the NIR intensity is well into the tunneling regime, ionization is not the only process that can occur. Some neutral population can survive the intense NIR field, resulting in an excited electronic (process 1 in Fig. 1) and vibrational population (process 2). Direct tunnel ionization of the valence electron will result in the population of the two spin-orbit split ground states in the molecular ion, $(e_{3/2})^{-1}\tilde{X}^2E_{3/2}$ and $(e_{1/2})^{-1}\tilde{X}^2E_{1/2}$ (process 3).

To probe the transient state of the $CH_3Br$ population, we tune an isolated attosecond XUV pulse[23] to the $M_{4,5}$ edge of Br corresponding to the excitation of the Br $3d$ electron. Since the binding energy of the Br $3d$ electron in $CH_3Br$ is 76.4 eV[24], the pre-edge corresponding to core-to-valence transitions is accessed in the energy range of 60–75 eV. A transient absorption spectrum is collected by referencing the $M_{4,5}$ transitions with and without the NIR-pump pulse. The quantity recorded corresponds to a change in optical density, or $\Delta O.D. = -\ln[I_{XUV+NIR}(E,\tau)/I_{XUV}(E)]$, where $I_{XUV}(E)$ is the attosecond pulse spectrum transmitted through the $CH_3Br$ gas cell, $I_{XUV+NIR}(E,\tau)$ is the transmitted attosecond pulse spectrum in the presence of the NIR pump pulse, $E$ is the XUV energy, and $\tau$ is the time delay between the NIR-pump and XUV-probe pulses. The transient absorption spectrum recorded at $\tau = 50$ fs is shown on the right side of Fig. 1 along with the spectrum, $I_{XUV}(E)$, of the isolated attosecond pulse (inset) used in the experiment.

Three main features are observed in the transient absorption spectrum corresponding to each of the strong-field initiated processes mentioned above. The lowest energy band below 65 eV (process 1) comprises a set of absorption lines arising from the excitation of a $3d$ electron to fill a valence hole in the $\sigma^\star$ excited neutral molecule. At the highest energy (process 2), we observe a broad negative absorption peak around 71 eV corresponding to the excitation of the $3d$ electrons to fill the lowest unoccupied molecular orbital (LUMO). The absorption occurs in the ground state of the molecule and is negative due to depletion of the ground state by the strong NIR field resulting in decreased absorption. The middle absorption band (process 3) is composed of a set of three absorption lines corresponding to the excitation of the $3d$ electron to fill the valence hole in the ionic ground state of $CH_3Br^+$.

To probe how these strong-field initiated processes evolve over time, we record the transient absorption spectrum as a function of time delay between the NIR-pump and XUV-probe fields. The resulting experimental ATAS trace is shown in Fig. 2a for a NIR pump intensity of $I = 1.5 \times 10^{14}$ W/cm², revealing rich time dependence in each absorption band. Further, the instrumental response time of the excited neutral absorption feature is measured to be $2.0 \pm 0.2$ fs [Fig. 2b], demonstrating the exquisite temporal resolution provided by the strong field, ATAS pump-probe experiment.

**Process 1: Passage through a conical intersection.** The energetic location of the absorption features we label as process 1 are delay dependent, converging to asymptotic values at large time delays, as shown in Fig. 3a. The asymptotic location of these absorption features arise from $M_{4,5}$ transitions in atomic Br at 65.0 eV [Br* ($P_{1/2} \rightarrow D_{3/2}$)], 64.4 eV [Br ($P_{3/2} \rightarrow D_{5/2}$)][25], and 64.1 eV [Br* ($P_{1/2} \rightarrow D_{5/2}$)], implying that molecular dissociation has taken place. While the transition at 64.1 eV is strictly dipole

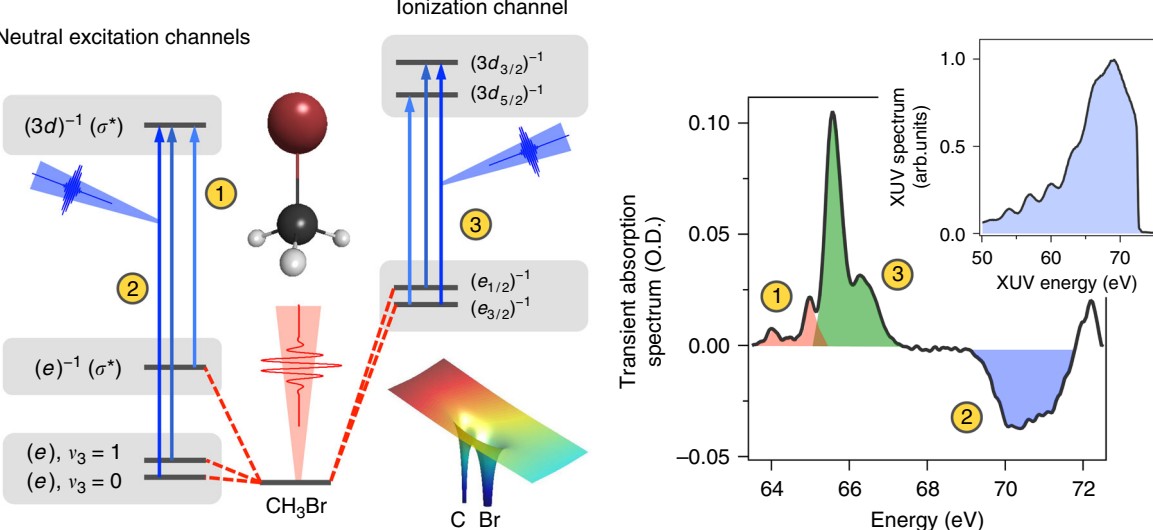

**Fig. 1** Attosecond Transient Absorption Spectroscopy (ATAS) Pump-Probe Scheme in $CH_3Br$. A 4 fs NIR pulse with an intensity of $I = 1.5 \times 10^{14}$ W/cm$^2$ is used to launch multiple valence wavepackets in $CH_3Br$, including (1) a neutral excited-state wavepacket, (2) a ground-state vibrational wavepacket, and (3) an ionic spin-orbit coupled wavepacket. The evolution of these wavepackets are probed using ATAS to follow the delay-dependent $M_{4,5}$ transition of Br, corresponding to the excitation of the Br 3d electron to various valence orbitals. The right-hand plot presents the transient absorption spectrum of $CH_3Br$ at a delay between the NIR-pump field and XUV-probe field of $\tau = 50$ fs. The spectroscopic signatures for each wavepacket can clearly be observed from this transient absorption plot. The isolated attosecond XUV spectrum is plotted in the inset as a reference

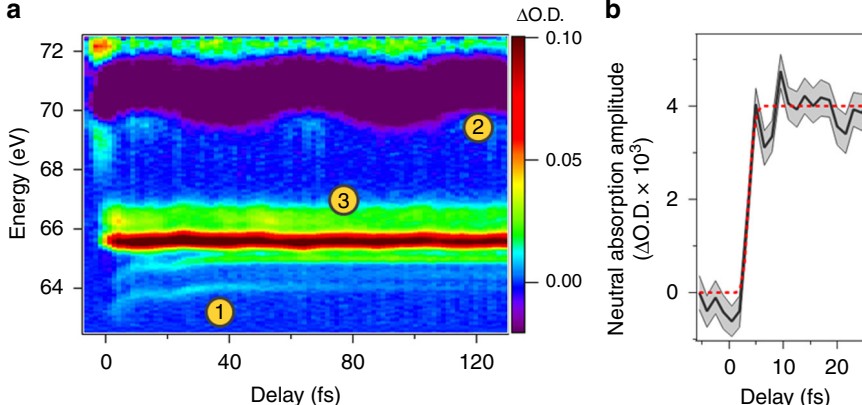

**Fig. 2** Experimental Attosecond Transient Absorption Spectroscopy (ATAS). **a** The delay-dependent ATAS trace labeling the time-dependent spectroscopic signatures for each valence wavepacket process listed in Fig. 1 [i.e., (1) a neutral excited-state wavepacket, (2) a ground-state vibrational wavepacket, and (3) an ionic spin-orbit coupled wavepacket]. **b** An energetic sum over process 1 (experimental sum over 62.5–65.2 eV, black curve), corresponding to the neutral excited state wavepacket, yields a corresponding instrumental response time of 2.0 ± 0.2 fs (red-dashed curve, corresponding fit). The shaded curve represents the measured uncertainty in Δ O.D

forbidden in the atomic limit due to spin-orbit selection rules, in the molecule, the Br atom can couple to an unpaired electron in the methyl group, lifting the transition restriction. Based on the location of the absorption lines at time zero, these atomic features are assigned to fragmentation from the neutral excited A-band of $CH_3Br$ corresponding to the excitation of a valence electron into a $\sigma^*$ molecular orbital.

From previous studies[20,26], it is known that the excited state spectrum of $CH_3Br$ is primarily composed of three states: $^3Q_1$, $^1Q_1$, and $^3Q_{0^+}$. The $^3Q_1$ and $^1Q_1$ states correlate to the Br $(P_{3/2})$ atomic limit while the $^3Q_{0^+}$ state correlates to the Br*$(P_{1/2})$ limit. The potential energy curves for these states are plotted in Fig. 3b. As can be seen, a conical intersection exists between the $^1Q_1$ and $^3Q_{0^+}$ states, which can lead to non-adiabatic population transfer between the two states during dissociation. Previous reports have used energy-resolved photofragments to measure both the direct

(via direct dissociation) and indirect (via non-adiabatic transitions) yield of Br/Br* upon photodissociation of these A-band states. The experimental reports estimated a non-adiabatic transition probability between $^3Q_{0^+}$ to $^1Q_1$, ranging from 0.14 to 0.38[20], 0.17[21], and 0.294[22]. The remaining photofragment yield is expected to arise from direct dissociation of the $^1Q_1$ and $^3Q_{0^+}$ to the Br and Br* atomic limits, respectively. As a result, in the red wing of the excitation spectrum, the two states have fairly equal excitation probabilities resulting in an equal branching ratio between Br and Br*. While in the blue wing of the excitation spectrum, the excitation is dominated by $^1Q_1$ resulting in a primary Br yield[26].

However, it is apparent from Fig. 3a that fragmentation into the Br* atomic limit dominates the dissociation dynamics in the present experiment, with very little population present in the Br atomic limit. In addition, within the experimental signal-to-noise

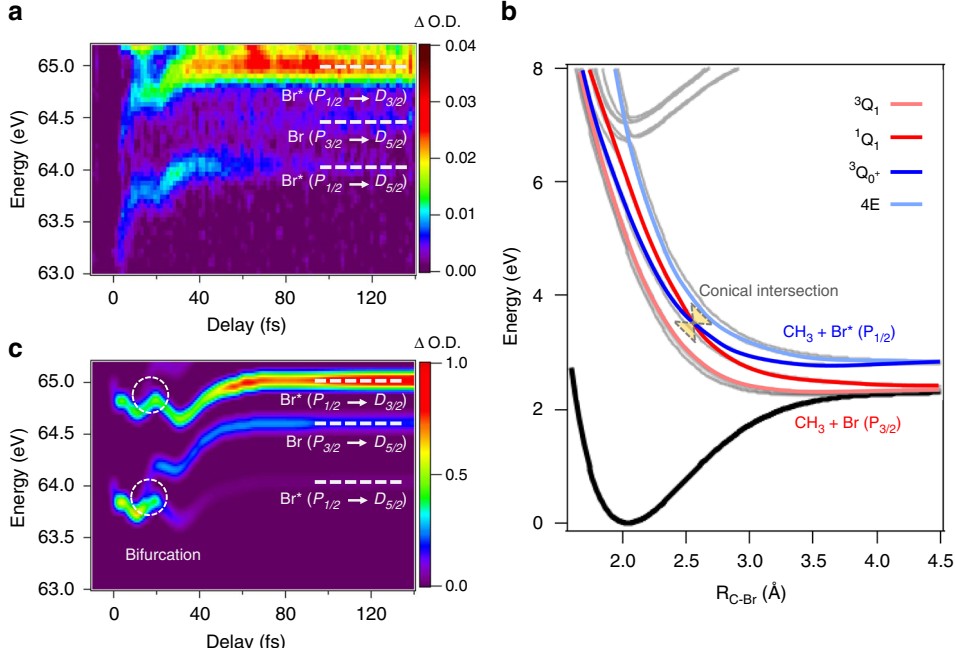

**Fig. 3** Neutral excited state dynamics in $CH_3Br$. **a** Attosecond transient absorption spectroscopy (ATAS) trace corresponding to the neutral excited state dynamics composing process 1. An additional transition, Br ($P_{3/2} \rightarrow D_{3/2}$) (65.4 eV), is predicted to be present, however is obscured in the present experiment due to the ionic absorption features above 65 eV (see Supplementary Note 1 and Supplementary Fig. 2). **b** A collection of ground, excited, and Rydberg state potential energy curves for $CH_3Br$ (computation described in Supplementary Note 3). The bold color curves represent the primary states that compose the excited state band. A conical intersection exists between the $^1Q_1$ and $^3Q_{0^+}$ excited states that can lead to non-adiabatic population transfer between the two dissociation pathways converging to Br and Br*. **c** Simulated ATAS dynamics for neutral excited states in $CH_3Br$. From the results of the simulation, we find that the ATAS dynamics are dominated by the $^3Q_{0^+}$ state, resulting in a dominant Br* atomic yield. A signature of the non-adiabatic transition can be seen as the emergence of Br population at greater time delays in **a**, **c**

ratio (SNR), only a single pair of spin-orbit, excited state absorption features is observed at the moment of neutral excitation. This suggests that the $^3Q_{0^+}$ state dictates the excitation dynamics. We therefore simulate the wavepacket dynamics assuming primary excitation to the $^3Q_{0^+}$ valence excited state and calculate the $M_{4,5}$ transitions at each time delay in order to reconstruct a simulated ATAS spectrogram. Details of the theoretical calculations, including the construction of a quasi-diabatic Hamiltonian of the system, quantum nonadiabatic dynamics simulations, and the simulation of ATAS trace are available in Supplementary Note 3. The simulated ATAS trace of the $^3Q_{0^+}$ excitation is presented in Fig. 3c, in good qualitative agreement with the experimental trace in Fig. 3a outside of the pump-probe overlap region. Within the overlap region, additional strong field effects should lead to Stark shifts in the absorption features which are not included in the present simulation.

From the simulated wavepacket dynamics(see Supplementary Note 3 and Supplementary Fig. 6) we find that ~20% of the $^3Q_{0^+}$ state undergoes a non-adiabatic transition through the conical intersection into the $^1Q_1$ state. The non-adiabatic transition leads to wavepacket bifurcation, resulting in a weak fragmentation into the Br atomic limit. This bifurcation is readily observed in the simulated ATAS spectrogram as a splitting in the molecular absorption feature into the different spin-orbit atomic limits at a time delay of 16 fs. Experimentally, we observe the growth of Br population at later time delays, providing a signature for the non-adiabatic population created by the conical intersection. However, due to the congestion of multiple line centers at early time delays and limited signal-to-noise, it is difficult to directly observe the bifurcation time experimentally. This quantity can be experimentally measured by subtracting out the two most prominent Br* transitions in the data and measuring the rise of Br yield (see

Supplementary Note 1 and Supplementary Fig. 3). Such an analysis yields a bifurcation time of $\tau_b = 15.0 \pm 0.4$ fs, in near agreement with the simulated result. Further, this measured bifurcation time is also in remarkable agreement with bifurcation time of 13 fs measured by Corrales et al.[9] for the same excited states probed in a different methyl halide molecule, $CH_3I$. While contributions from the excitation and direct dissociation of the $^1Q_1$ state cannot be unambiguously ruled out in the Br atomic yield, the remarkable agreement with the simulated dynamics and prior work strongly suggests that the wavepacket dynamics are directly associated with the presence of the conical intersection. Further experiments with increased energy resolution and better SNR should prove to be illuminating in directly resolving these weak non-adiabatic transitions in the time domain without post-processing.

The contrasting excitation behavior in the present experiment is suspected to arise due to a strong field depletion mechanism, characterized by a Keldysh parameter of $\gamma \approx 0.8$. It is known from previous strong field ionization experiments[27] that $CH_3Br$ is preferentially ionized perpendicular to the C–Br axis. Therefore, excitation to both the Rydberg and ionization continuum dominate perpendicular transitions and greatly deplete the population available for excited states prepared perpendicular to the molecular axis, i.e. the $^3Q_1$ and $^1Q_1$ states. However, the parallel transition to $^3Q_{0^+}$ is largely unaffected by this strong field depletion mechanism. Further, we find that the calculated $M_{4,5}$ transitions in the Rydberg states (arising from perpendicular excitations) are substantially weaker than in the $^3Q_{0^+}$ state. Therefore, the $^3Q_{0^+}$ state is the most visible state dictating the neutral excited state dynamics in the ATAS experiment.

Finally, it is important to note that the energy dependence of these neutral features is not a direct measure of the excited state

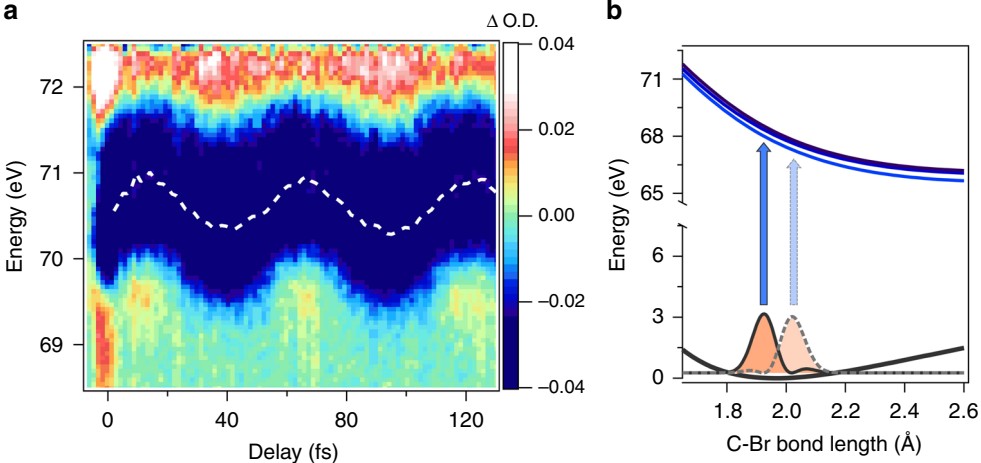

**Fig. 4** Ground State Vibrational Wavepacket in $CH_3Br$. **a** Zoom into the portion of the attosecond transient absorption trace depicting the ground-state vibrational wavepacket from process 2. A clear center-of-energy oscillation is visible in the transient absorption spectrum (vibrational period of $T = 53.9 \pm 0.2$ fs) due to the wavepacket motion of the $R_{C-Br}$ bond. **b** Qualitative picture demonstrating how the time dependence of the $R_{C-Br}$ bond can trace out different $M_{4,5}$ transition energies

dynamics. Instead it is a measure of the slight differences in potential energies between the initial and final states in the core-to-valence transitions. From the simulation we find that the transient change in absorption energy observed within the first 40 fs of the experiment actually arises due to a potential well located in the core-excited state potential energy surface. However, the intensity of the neutral absorption features can serve as a direct measurement of the excited state dynamics. Since the spin-orbit transition at 64.1 eV is forbidden in the atomic limit, the intensity of this band provides a model-independent measurement of the photo-dissociation time of the $^3Q_{0^+}$ state, which we measure to be $\tau = 68 \pm 3$ fs (see Supplementary Note 1 and Supplementary Fig. 4). This number falls within the dissociation time upper limit previously estimated by Gougousi et al.[20] to be $\tau_{UL} = 120 \pm 40$ fs.

**Process 2: Ground state vibrational dynamics**. Next, we consider the absorption feature representing process 2 in Figs. 1 and 2. An enlarged view of the transient absorption trace focusing on process 2 is shown in Fig. 4a. The negative absorption peaks depict a clear center-of-energy oscillation as a function of time delay, with an oscillation period of $T = 53.9 \pm 0.2$ fs. This oscillation period agrees well with the C–Br vibrational period[28] in the neutral ground state of $CH_3Br$ (54.6 fs, or 611 cm$^{-1}$, corresponding to the $\nu_3$ vibrational eigenmode), suggesting that these dynamics arise due to a ground state vibrational wavepacket. A coherent superposition between several $\nu_3$ vibrational levels in the $CH_3Br$ ground state will result in a time-dependent C–Br bond length, or $R_{C-Br}(t)$. Since the absorption peaks in Process 2 correspond to an excitation to the final state $(3d)^{-1}(\sigma^*)$ exhibiting repulsive character with respect to $R_{C-Br}$, the non-stationary bond length will trace out different $M_{4,5}$ absorption energies as a function of time delay, as displayed in Fig. 4b. Therefore, these energetic shifts in the ground state absorption features sensitively portray vibrational structure and motion in the electronic ground state.

A ground state vibrational wavepacket can be prepared through a few possible strong-field mechanisms, including both thermal (e.g., Lochfraß[29]) and non-thermal (e.g., bond-softening[30] and stimulated Raman) mechanisms. Due to the low thermal population in the excited $\nu_3$ vibrational states and the measured excitation phase of the wavepacket, $\phi = (0.52 \pm 0.02)\pi$ (sine wave), the wavepacket is most likely prepared by a non-

thermal process such as the bond-softening mechanism detailed by Wei et al.[30] (more detail in Supplementary Note 2).

**Process 3: Coherent spin-orbit wavepacket in $CH_3Br^+$**. As mentioned above, the absorption lines attributed to process 3 arise due to tunnel ionization of neutral $CH_3Br$. The process of tunnel ionization is capable of preparing a mixed state composed of multiple ionic states[14,31,32]. Here, we ionize to the pair of spin-orbit states in the ground state configuration of the molecular ion, resulting in the three transitions: L1, $(e_{3/2})^{-1} \rightarrow (3d_{5/2})^{-1}$ (65.6 eV); L2, $(e_{1/2})^{-1} \rightarrow (3d_{3/2})^{-1}$ (66.2 eV); L3, $(e_{3/2})^{-1} \rightarrow (3d_{3/2})^{-1}$ (66.6 eV). While ligand field splittings can give rise to additional sub-peaks in these absorption features[33], they are not discernible in the present experiment. Gaussian fits to the L1 (red), L2 (orange), and L3 (green) absorption spectra are shown in Fig. 5a. Previous reports have demonstrated that a mixed state prepared by tunnel ionization can exhibit a measurable degree of coherence in atomic systems when the ionizing field fulfills one of two conditions[14,34]: (i) the pulse duration of the ionizing field is less than the electronic period of the mixed state, $T_{12} = 2\pi\hbar/E_{12}$, where $E_{12}$ is the energy splitting between the two levels composing the mixed state[34] or (ii) $E_{12}$ overlaps with a harmonic of the ionizing field frequency[35]. The energy level splitting of the ionic spin-orbit split states in $CH_3Br$ is $E_{12} = 320$ meV corresponding to a period of $T_{12} = 12.9$ fs. Since the driving pulse duration in the present experiment is well below $T_{12}$, the generation of a coherent spin-orbit wavepacket in the molecular ion is readily feasible. Further, provided that excitation of this coherent pair of spin-orbit ionic states arrives in the same final core-hole excited state upon XUV excitation [i.e., $(3d_{3/2})^{-1}$], a modulation of the XUV transition probability will reveal the wavepacket dynamics.

The linewidth of each individual spin-orbit absorption feature is measured to be approximately 450 meV, leading to spectral blurring of the absorption lines. This makes the direct observation of a spin-orbit oscillation difficult to monitor. Therefore, in order to measure the spin-orbit wavepacket, a sum is performed over the L1, L2, and L3 ionic transitions in $CH_3Br$ in Fig. 2a as a function of strong field pump, attosecond-probe time delay. The coherent spin-orbit wavepacket is defined by an off-diagonal, complex coherence term in the ion density matrix. This coherence term is imprinted on a complex lineshape function giving rise to both an absorptive and dispersive lineshape

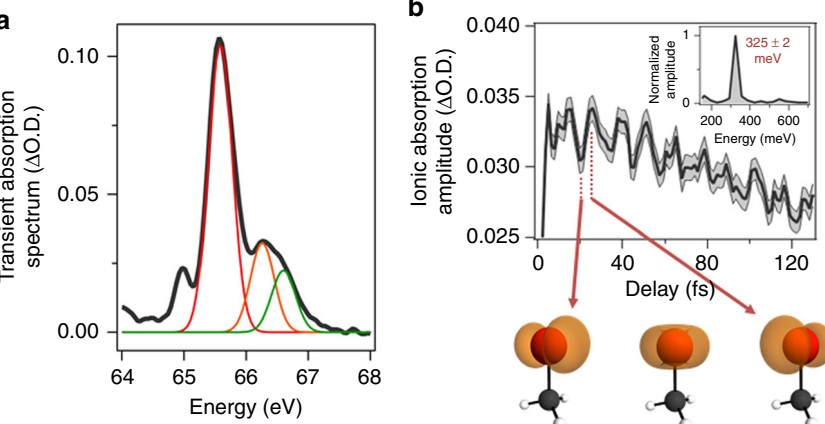

**Fig. 5** Observation of a Spin-Orbit Electronic Wavepacket in $CH_3Br^+$. **a** Spectral line out of the attosecond transient absorption trace depicting the composition of the $CH_3Br$ ionic absorption profiles corresponding to process 3. This absorption profile is composed of three spin-orbit transitions labeled as L1 (red), L2 (orange), and L3 (green) in the manuscript. **b** By summing over the three spin-orbit transitions, we can uniquely measure the spin-orbit wavepacket dynamics generated by the strong field ionization process. The shaded curve represents the measured uncertainty in $\Delta$ O.D. A Fourier analysis of these dynamics (inset) clearly illustrates a spin-orbit wavepacket with a frequency of $325 \pm 2$ meV, corresponding to a period of 12.7 fs. The qualitative picture for this process is presented by the electron hole density screen shots for $CH_3Br^+$ depicted below **b**, corresponding to the oscillation in electron hole density between $p_x$ and $p_y$ character about the C–Br axis

exhibiting contrasting dynamics with respect to the coherence term[36]. As a result, the absorptive term (on-resonance) for a contrasting pair of spin-orbit transitions oscillates in phase while the dispersive term (off-resonance) oscillates out of phase. By summing over all spin-orbit transitions, the in-phase contributions of the absorption features are coherently averaged to reveal the spin-orbit wavepacket dynamics shown in Fig. 5b. This sum exhibits a pronounced oscillation corresponding to a beating frequency of $325\pm$ meV, in good agreement with the predicted spin-orbit splitting of $CH_3Br^+$. The qualitative picture for the electron hole dynamics in such a spin-orbit wavepacket is shown below Fig. 5b, corresponding to the oscillation in electron hole density between $p_x$ and $p_y$ character about the Br atom (i.e., $e_{3/2}$ and $e_{1/2}$).

## Discussion

We have demonstrated the utility of ATAS in probing non-adiabatic transitions arising due to a conical intersection during the photo-fragmentation of excited $CH_3Br$. The superb resolution provided by the strong field excitation and attosecond probe is essential for resolving these weak non-adiabatic signatures. Future experiments with increased spectral resolution and SNR should prove to be illuminating for directly probing the bifurcation of an excited-state wavepacket due to these non-adiabatic intersections. In addition, we resolved competing strong field initiated dynamics giving rise to coherent vibrational and electronic motion in both the neutral and molecular ion respectively. The ability to simultaneously probe these disparate and intricate molecular dynamics reflects the potential of ATAS as a general tool to resolve valence reactions in a host of interesting chemical and bio-chemical systems, with timescales ranging from attoseconds to nanoseconds. Finally, the extension of this technique to the soft X-ray regime[37–39] should allow the simultaneous probing of oxygen, nitrogen, and carbon edges, resulting in unprecedented ultrafast studies of organic matter at the attosecond timescale.

## Methods

**Experimental set-up**. The experimental set-up for the generation of a few-cycle, NIR pump pulse and the isolation of an XUV, attosecond probe pulse is detailed in ref.[23]. Briefly, femtosecond pulses are generated from a carrier-envelope phase (CEP) stable Ti:Sapphire oscillator and amplified to a pulse energy of 2 mJ at 1 kHz.

The CEP is stabilized to ~100 mrad within the amplifier at a pulse duration of 27 fs. The CEP-stable pulses undergo nonlinear, spectral broadening within a Ne-filled stretched hollow-core fiber and are then compressed using both a set of broadband chirped, multi-layer mirrors to compensate for the second-order dispersion and a 1 mm thick ammonium dihydrogen phosphide (ADP) crystal to compensate for the third-order dispersion after non-linear broadening, yielding a compressed pulse duration <4 fs.

For the pump-probe experiment, the few-cycle pulses are divided into a pump and probe arm using a 50:50 beamsplitter. The NIR-pump pulse travels through a delay stage and is focused into a 2 mm path length gas cell filled with 10 torr of $CH_3Br$. The NIR beam is focused to a beam radius of $w_o = 70$ μm, resulting in a peak intensity of $I = 1.5 \times 10^{14}$ W/cm², sufficient to launch multiple strong-field initiated dynamics. The XUV-probe pulse is generated by focusing the few-cycle NIR pulse in the probe arm into a 2 mm path length gas cell backed with ~30 torr of Ar. Through the process of HHG, the NIR light is up-converted into higher-order harmonics in the XUV energy regime. Single attosecond pulses are isolated from the XUV spectrum by employing the amplitude gating technique beyond the Cooper minimum of Ar[23]. The residual NIR light is filtered out of the probe arm with a 200 nm thick Al filter. A toroidal mirror is then used to focus the isolated attosecond pulse into the $CH_3Br$ gas cell where it probes the excited state molecular ensemble. The delay between the NIR-pump and XUV-probe arms is actively stabilized to <100 as using an in-line interferometer at 405 nm. Finally, the pump NIR field is removed from the beam path after the $CH_3Br$ gas cell using a second 200 nm thick Al filter. The attosecond pulse spectrum is spectrally dispersed with a gold coated, XUV grating and measured with an X-ray camera. A beam shutter is programmed to block the NIR pump arm every other shot in order to reference the $CH_3Br$ XUV absorption spectrum with and without the NIR pump pulse and measure the change in optical density, $\Delta$O.D.

The average single-shot, standard deviation in $\Delta$O.D. across the ATAS spectrum (60–72 eV) is $\sigma_{O.D.} = 0.07$, in good agreement with the uncertainty of $\sigma_{O.D.} = \sqrt{2}(\Delta I/I) = 0.07$ calculated from the 5% fluctuation experimentally measured in the XUV amplitude. Therefore, in order to reduce the uncertainty, the transient absorption spectrum is averaged over 200 frames, resulting in an average standard deviation of $\sigma_{O.D.} = 0.005$. However, this still accounts for a significant fraction of the transient absorption amplitude in the neutral excited state portion of the spectrum. One of the major sources of noise in the present experiment is the CEP instability of the few-cycle NIR pulses, which can give rise to (i) center-of-energy jitter and (ii) amplitude fluctuations in the isolated attosecond pulse spectrum. This CEP noise will translate into both a (i) linear and (ii) DC background in the transient absorption spectrum, respectively. To remove the contribution of this CEP noise, a linear background is subtracted from the transient absorption spectrum at each time delay between two spectral points that are expected to exhibit zero absorption signal, specifically 62 and 67.4 eV. An example of this analysis is presented in Supplementary Fig. 1 comparing the raw transient absorption spectrogram against the transient absorption spectrogram after the linear subtraction.

**Simulation details**. The potential energy surfaces for valence and Rydberg states of neutral $CH_3Br$ were calculated using State-Averaged Complete Active Space Self-Consistent Field (SA-CASSCF) theory using Molpro[40](see Supplementary Note 3

and Supplementary Fig. 5). The active space consisted of Br p lone pair orbitals, C–Br σ bonding and antibonding orbitals and the 5 s Rydberg orbital. The aug-cc-pVTZ-pp basis set and the associated Relativistic Effective Core Potential (RECP)[41] were used for Br and the aug-cc-pVTZ basis for all other atoms. Additional diffuse functions were added to improve the description of Rydberg states. Spin-orbit couplings were obtained from the RECP. Analytic coupled quasi-diabatic potentials were constructed along the C–Br dissociation coordinate from a grid of ab initio data points. Diabatization was achieved by diagonalizing the $\hat{R}^2$ quadrupole operator within each symmetry block[42]. Nonadiabatic quantum dynamics simulations were performed on the 1-D coupled potentials using split-operator fourier transform[43] method with 4096 grid points spanning from 2.0 a.u. to 20.0 a.u (see Supplementary Fig. 6). In each simulation, the ground vibrational wave packet of the ground electronic state was lifted to a different excited-state and propagated for 169 fs using time steps of 0.048 fs.

The core excitation process was calculated at Restricted-Active-Space Configuration Interaction (RASCI) level of theory, also using Molpro. The orbitals were obtained from SA-CASSCF calculation of the valence states and during the final configuration interaction step the total occupation of 3d orbitals was restricted to 10 for the valence states and 9 for the core-excited states. Spin-orbit couplings were obtained from RECP on Br. A total of 12 valence and 20 core-excited spin-orbit states were calculated in the RASCI calculations, along with transition dipole moments between all pairs of states. Excitation energies and transition dipoles on a grid along the 1-D reaction coordinate were interpolated to yield analytic functions. These were then combined with the time-dependent densities obtained in the nonadiabatic dynamics simulations to generate the simulated ATAS signals of the dominant $^3Q_{0+}$ state in 3 (computational details described in Supplementary Note 4 and combined signals of all initial excitations in Supplementary Fig. 2).

## Data availability
The data that support the findings of this study are available from the corresponding author upon reasonable request.

## Code availability
The codes used to simulate the wavepacket and ATAS dynamics are available from the corresponding authors upon request.

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

## Acknowledgements
H.T., Y.K., D.M.N., and S.R.L. were supported by the Army Research Office (ARO) (W911NF-14-1-0383). X.Z. and T.J.M. acknowledge support by the AMOS program within the Chemical Sciences, Geosciences and Biosciences Division of the Office of Basic Energy Sciences, Office of Science, U.S. Department of Energy. M.S. and M.R. were supported by the National Science Foundation (NSF) (CHE-1361226 and CHE-1660417). Z.L.

acknowledges partial financial support from the VW foundation through Peter Paul Ewald fellowship. Y.K. additionally acknowledges financial support from the Funai Overseas Scholarship. We would like to thank Andrew Attar and Aditi Bhattacherjee for their help in setting up a vacuum integrated liquid nitrogen trap for handling halogenated species as well as Stefan Pabst for his fruitful discussions on strong field excitation mechanisms. We would also like to thank Rolf Heilemann and Henrik Koch for their help in CVS-CC method.

## Author contributions

H.T., M.S., D.M.N. and S.R.L. designed the experiment. H.T., Y.K., M.S. and M.R. performed the experimental measurements. X.Z., Z.L., M.H. and T.J.M. constructed the coupled potential energy surfaces and performed the wavepacket and ATAS simulations. H.T., X.Z., Z.L., Y.K., T.J.M., D.N.M., and S.R.L. wrote the paper. All authors discussed the results and contributed to the paper.

## Additional information

**Competing interests:** The authors declare no competing interests.

