## [Peer Review File · Nature Communications]

Reviewers' comments:

Reviewer #1 (Remarks to the Author):

This is an impressive paper on the time-resolved photodynamics of methyl bromide induced by strong (1.5×10^{14} W/cm²) and ultrashort (4 fs) NIR pulses and probed by the 3d core-to-valence transition using attosecond transient absorption. Isolated attosecond pulses generated by high harmonic generation and 4 fs NIR pulses provide an impressive time-resolution of 2.0 ± 0.2 fs, which is excellent to disentangle the different ultrafast photodynamics induced in methyl bromide by the strong NIR pulse. The authors report three different photodynamics: the passage through a conical intersection in valence excited states, the creation of a vibrational wave packet in the ground state and the creation of a coherent spin-orbit wavepacket in the ground state of the cation. In addition, theoretical calculations including high level ab initio potential energy curves of the relevant excited states, diabaticization and quantum wavepacket calculations along with simulations of the ATAS spectra have been performed to help interpret the experimental results.

The paper is well written, the discussions are supported by the experimental data and the theoretical calculations help to interpret the measurements. The information contained in the supplementary material is very useful. I recommend publication in Nature Comm. and only suggest some minor revisions along the following comments:

1.- I find the title too constrained to one of the photodynamics studied while the experiments are so rich that there are other two photodynamics involved and disentangled. I suggest to change the title of the paper to include all the dynamics observed. My suggestion is "Probing conical intersections and vibrational and spin-orbit wavepackets with attosecond transient absorption spectroscopy".

2.- The authors say in the introduction, at the end of 3rd paragraph, "However, no study has directly probed these non-adiabatic dynamics in real time". I do not agree with this statement since there are at least three recent papers on the subject on similar molecules revealing the bifurcation in real time, which should be included in the references list and briefly mentioned in the introduction:

- Yang et al., Imaging CF₃I conical intersection and photodissociation dynamics with ultrafast electron diffraction, *Science*, 361, 64 (2018).

- Allum et al., Coulomb explosion imaging of CH₃I and CH₂ClI photodissociation dynamics, *J. Chem. Phys.* 149, 204313 (2018).

- Corrales et al., Coulomb explosion imaging for the visualization of a conical intersection, *J. Phys. Chem. Lett.* 10, 138 (2019).

3.- Regarding process 1:

a) Have the authors considered possible strong field effects, such as Stark shift or generation of doubly charge species and Coulomb explosion, on the dynamics when using such an ultrashort strong NIR pulse (1.5×10^{14} W/cm²)? The authors should comment on this in the manuscript.

b) Taking into account the small discrepancy in Fig. 2 between experiment and theory on the relative population of the lower Br* limit and the Br one, might the authors consider the possible role of minor excitation to the 1Q1, including a "reverse" transfer of population 1Q1 -> 3Q0?

c) There is a problem with labels in Figure 3. According to the figure caption, label (b) must be (c) and viceversa. In addition, the dash circles depicted in panel (b) must be explained in the figure caption. Bifurcation should be mentioned in the figure caption.

d) The authors report a bifurcation time of 15.0 ± 0.4 fs for CH₃Br. I find this time in very good agreement with that reported in the work of Corrales et al. , J. Phys. Chem. Lett. 10, 138 (2019) for CH₃I of 13 fs and therefore I suggest the authors to mention it in the text.

d) The sentence "Further experiments with increased energy resolutions and better signal-to-noise... without post-processing" should be included in the Conclusions.

c) I do not think the use of "lifetime" to represent the dissociation time is appropriate for a fast photodissociation process. I would suggest the use of "dissociation time" instead. Actually, the comparison with the "lifetime" reported by Gougousi et al. is not appropriate since they did not measure the reaction time but they estimate it from the fragment angular distribution using nanosecond lasers. This must be clarified in the text.

e) How the oscillations in energy observed on the simulated ATAS at short times are explained, besides the bifurcation, which is clearly observed and indicated?

4.- I assume that the potential energy curves depicted in Fig. 2c have been computed in the present work. If so, it should be mentioned in the figure caption and refer here to the supplementary material.

5.- Regarding process 2. The sentence in page 5 "Since the absorption peaks in Process 1 correspond to an excitation ... , as displayed in Fig. 4 (b).", appears a little unclear to me. Which is the relation with Process 1?

6.- A small typo in page 6 at the beginning of the second paragraph where I guess "moniter" should be replaced by "monitor".

Reviewer #2 (Remarks to the Author):

Title: Probing conical intersection dynamics with attosecond transient absorption spectroscopy

This manuscript describes what is presented as a attosecond transient absorption spectroscopic (ATAS) experiment in which the pump pulse is a strong-field 4fs NIR pulse, which prepares a complex wave packet (in the molecule CH₃Br) comprised of ground electronic but vibrationally excited states, electronically excited states, as well as multiple ionic continua. This wave packet is subsequently probed with a time-delayed XUV pulse that excites a 3d electron on the Br atom into the manifold of virtual orbitals (IP ~76 eV). The resulting time-evolution of the transient absorption spectrum is then analyzed to extract information about the initially prepared wave packet.

This work clearly shows the utility of using core spectroscopies to probe complex multi-state molecular dynamics. Each of the components of initially prepared wave packet are differentially projected out in the energy resolved spectrum. The NIR pump pulse will, in general, create a very complicated wave packet comprised of numerous electronic and vibrational components, which makes subsequent analysis of some time-resolved signal highly problematic. In this case, it is precisely that characteristic of strong-field NIR excitation pulses that showcases the ability of the XUV pulse to disentangle and energy resolve all the different wave packet components.

However, there are a couple of issues with the manuscript as currently presented.

Firstly, the term ATAS is inappropriate as the dynamics they study is on the femtosecond time scale (as shown in their Figs). Their time resolution is also in the femtosecond, NOT sub-femtosecond (attosecond) time scale (as shown by their data). They should call it "ultrafast", not attosecond transient absorption.

Secondly, one notes that the experimental results are least compelling when it comes to the process that gives rise to the title of the manuscript: describing the passage through the conical intersection. The simulated spectrum clearly shows the signature of a bifurcation in the wave packet corresponding to an electronic transition, but the experimental spectrum does not. Rather, this passage is inferred from a delayed increase in a different dissociation channel -- although it's not obvious that a conical intersection is the sole explanation for that observation (i.e. dissociation resulting from direct preparation of the 1Q1 state). Furthermore, the supplementary material seems

to suggest that this spectrum shown in Figure 3c was generated assuming that only 3Q0+ state was populated via the NIR pulse. This assumption makes the bifurcation of wave packet most clear in the figure, but is unlikely to rigorously hold. Confirming the validity of this assumption is possible via simulations of the NIR pump process, but they would be non-trivial.

Furthermore, They do not analyze the structures "during" the passage through the conical intersection. Their data show the 'before' and the 'after', not so different from the many other studies of internal conversion in molecules. The time scale may be faster here, but the result is the same: they see the 'before' and the 'after'.

In comparison, the assignments of the two other processes discussed are less ambiguous. The observation of the ground state vibrational wave packet and, in particular, the beats between the different ionic continua is relatively compelling. In general, the wealth of spectroscopic data amenable to assignment to the various components of the wave packet is impressive.

In summary, the present work is first rate and merits publication in Nature Communications. However, the title of the manuscript focuses specifically on the one part of the experiment/analysis that is somewhat ambiguous. The nonadiabatic dynamics is inferred largely on the basis of numerical simulation and subsequent comparison to the experimental spectrum -- where former assumes that the prepared wave packet is simplified version of what is likely generated via the strong-field pump pulse.

That said, the attosecond XUV probe is shown to be a highly effective approach to simultaneously image multiple components of a complex molecular wave packet. These methods are leading edge and will be of general interest to the chemical physics community.

Specific Comments

1. In Figure 3b: the curve coloring implies adiabatic states, but the legend implies diabatic states. Also: the labels (b) and (c) should be switched.

2. Some indication as to the nature/origin of simulated spectrum is warranted in the main body of the text.

3. The discussion of the formation of a coherent wave packet comprised of ionic states is somewhat terse and difficult to understand. I suggest the authors more clearly state: i) what N-photon paths would be expected to interfere to generate the beat frequencies? ii) what level spacings would the present observable be sensitive to? iii) what specifically, is the origin of the single beat observed in the spectrum?

Response to Reviewer #1:

This is an impressive paper on the time-resolved photodynamics of methyl bromide induced by strong (1.5×10^{14} W/cm²) and ultrashort (4 fs) NIR pulses and probed by the 3d core-to-valence transition using attosecond transient absorption. Isolated attosecond pulses generated by high harmonic generation and 4 fs NIR pulses provide an impressive time-resolution of 2.0 ± 0.2 fs, which is excellent to disentangle the different ultrafast photodynamics induced in methyl bromide by the strong NIR pulse. The authors report three different photodynamics: the passage through a conical intersection in valence excited states, the creation of a vibrational wave packet in the ground state and the creation of a coherent spin-orbit wavepacket in the ground state of the cation. In addition, theoretical calculations including high level ab initio potential energy curves of the relevant excited states, diabaticization and quantum wavepacket calculations along with simulations of the ATAS spectra have been performed to help interpret the experimental results.

The paper is well written, the discussions are supported by the experimental data and the theoretical calculations help to interpret the measurements. The information contained in the supplementary material is very useful. I recommend publication in Nature Comm. and only suggest some minor revisions along the following comments:

We thank the reviewer for their positive feedback and address the comments below.

1.- I find the title too constrained to one of the photodynamics studied while the experiments are so rich that there are other two photodynamics involved and disentangled. I suggest to change the title of the paper to include all the dynamics observed. My suggestion is “Probing conical intersections and vibrational and spin-orbit wavepackets with attosecond transient absorption spectroscopy”.

We agree with the reviewer’s opinion of the title as presenting a limited scope for the rich physics happening in the experiment and have decided to **change the title to, “Disentangling conical intersection and coherent molecular dynamics in methyl bromide with attosecond transient absorption spectroscopy.”**

2.- The authors say in the introduction, at the end of 3rd paragraph, “However, no study has directly probed these non-adiabatic dynamics in real time”. I do not agree with this statement since there are at least three recent papers on the subject on similar molecules revealing the bifurcation in real time, which should be included in the references list and briefly mentioned in the introduction:

- Yang et al., Imaging CF₃I conical intersection and photodissociation dynamics with ultrafast electron diffraction, Science, 361, 64 (2018).
- Allum et al., Coulomb explosion imaging of CH₃I and CH₂Cl₂ photodissociation dynamics, J. Chem. Phys. 149, 204313 (2018).
- Corrales et al., Coulomb explosion imaging for the visualization of a conical intersection, J. Phys. Chem. Lett. 10, 138 (2019).

We agree with the reviewer's assessment that there has been other interesting research in recent years regarding the direct probing of bifurcation in various methyl halides (e.g. Yang et al. and Corrales et al.). However, the original statement was correct as the sentence was referring specifically to methyl bromide. **The two papers noted by the reviewer (Yang et al. and Corrales et al.) are relevant citations and have been added into the introduction of the paper on pg. 1, paragraph 1. Furthermore, we have added a reference to Corrales et al. in our discussion of the bifurcation time (as requested below).**

3.- Regarding process 1:

a) Have the authors considered possible strong field effects, such as Stark shift or generation of doubly charge species and Coulomb explosion, on the dynamics when using such an ultrashort strong NIR pulse (1.5×10^{14} W/cm²)? The authors should comment on this in the manuscript.

Stark shifts should only occur when the pump and probe fields overlap in time. Therefore, we have not taken into account Stark shifts in the transient absorption model for process 1. The primary purpose of the wavepacket simulations is to interpret the dynamics that occur after excitation (> 10 fs for process 1) as the wavepacket is dissociating through the conical intersection. **However, this may not be clear to readers, so we have modified and included the sentences, "The simulated ATAS trace of the $^3Q_{0+}$ excitation is presented in Fig. 3 (c), in good qualitative agreement with the experimental trace in Fig. 3 (a) outside of the pump-probe overlap region. Within the overlap region, additional strong field effects should lead to Stark shifts in the absorption features that are not included in the present simulation" (pg. 4, paragraph 2).**

Regarding Coulomb explosion, this process will primarily occur in the dication, resulting in the fragmentation pathway $CH_3^+ + Br^+$. While the XUV probe can properly detect this pathway, we observe no absorption features corresponding to the core-valence absorption of a dicationic state. Further, the core-valence absorption of Br^+ occurs at higher energy ($\sim 66 - 67.5$ eV), and therefore does not affect the analysis of the neutral excited states in process 1. If Coulomb explosion is occurring in the experiment, resulting in the production of Br^+ , it could act as a DC contaminant of the ionic absorption features corresponding to process 3 – however, it should not impact the observation of the AC spin-orbit oscillation observed in process 3.

b) Taking into account the small discrepancy in Fig. 2 between experiment and theory on the relative population of the lower Br^* limit and the Br one, might the authors consider the possible role of minor excitation to the $1Q_1$, including a "reverse" transfer of population $1Q_1 \rightarrow 3Q_0$?

While we cannot unambiguously rule out a weak excitation to the 1Q_1 state, we can make two observations: 1) If we take a lineout of the transient absorption spectrum during the excitation of the neutral excited states (0.5 – 3.5 fs, plotted below; red-dashed curve represents a Gaussian fit to the data), we observe a spin-orbit pair of absorption lines ($\Delta E_{SO} = 1$ eV, corresponding to the splitting of the core excited state) which we assign to $^3Q_{0+} - D_{5/2}$ and $^3Q_{0+} - D_{3/2}$ with a linewidth of ~ 450 meV (corresponding to the width of the excited state wavepacket). The transient absorption simulation suggests that excitation to 1Q_1 should appear as a pair of transitions ~ 0.4 eV below this $^3Q_{0+}$ pair. However, no additional pair of absorption lines is

observed in the experimental lineout. Therefore, within the limit of the experimental signal-to-noise ratio (SNR), we do not observe another excited state in the initial excitation step. 2) We know from previous experiments that the molecule is preferentially ionized along the perpendicular direction of the C-Br bond. At an intensity of $1.5 \times 10^{14} \text{ W/cm}^2$, we are most likely depleting the available neutral population along this perpendicular axis. Therefore, the only states left to excite are the neutral parallel states. This is in line with observation (1), suggesting that $^3Q_{0+}$ is the predominant neutral excited state in the present manuscript. The discrepancy between experiment and theory observed at early time delays most likely arises from small errors in the computation of the core excited potential energy surfaces used to calculate the transient absorption energy. While point (2) is already discussed in the manuscript, we have not included any discussion of the fact that we only see a single pair of excited state absorption lines. **Therefore, we have added the following sentence to the manuscript, “In addition, within the experimental signal-to-noise ratio (SNR), only a single pair of spin-orbit, excited state absorption features is observed at the moment of neutral excitation” (pg. 4, paragraph 2).**

c) There is a problem with labels in Figure 3. According to the figure caption, label (b) must be (c) and viceversa. In addition, the dash circles depicted in panel (b) must be explained in the figure caption. Bifurcation should be mentioned in the figure caption.

We thank the reviewer for pointing out this issue and **we have corrected the ordering of (b) and (c) in the caption of Figure 3.**

d) The authors report a bifurcation time of $15.0 \pm 0.4 \text{ fs}$ for CH_3Br . I find this time in very good agreement with that reported in the work of Corrales et al. , *J. Phys. Chem. Lett.* 10, 138 (2019) for CH_3I of 13 fs and therefore I suggest the authors to mention it in the text.

Given the similarity in the two molecules, we agree that it is important to remark on the agreement between the two experiments. **We have included a sentence stating, “Further, this**

measured bifurcation time is also in remarkable agreement with bifurcation time of 13 fs measured by Corrales *et al.* (Corrales citation) for the same excited states probed in a different methyl halide molecule, CH₃I” (pg. 5, paragraph 1).

d) The sentence “Further experiments with increased energy resolutions and better signal-to-noise... without post-processing” should be included in the Conclusions.

While such a sentence exists in the conclusion already, **we have expanded that sentence to say, “Future experiments with increased spectral resolution and SNR should prove to be illuminating for directly probing the bifurcation of an excited-state wavepacket due to these non-adiabatic intersections” (pg. 6, paragraph 3).**

c) I do not think the use of “lifetime” to represent the dissociation time is appropriate for a fast photodissociation process. I would suggest the use of “dissociation time” instead. Actually, the comparison with the “lifetime” reported by Gougousi *et al.* is not appropriate since they did not measure the reaction time but they estimate it from the fragment angular distribution using nanosecond lasers. This must be clarified in the text.

We have changed “photo-dissociation lifetime” to “photo-dissociation time” in the manuscript and supplementary information (pg. 5, paragraph 3 in manuscript and pg. 4, paragraph 3 as well as caption of Fig. 5 in supplement). Regarding the comparison to Gougousi *et al.*, we have already noted that their measurement time serves as an upper limit for the photo-dissociation time of methyl bromide and that our number falls within this upper limit. Therefore, we think this statement holds true.

e) How the oscillations in energy observed on the simulated ATAS at short times are explained, besides the bifurcation, which is clearly observed and indicated?

Regarding the oscillation in the neutral excited absorption features, we have noted the origin for their delay dependence in the final paragraph of the section on process 1 in the original manuscript – *“it is important to note that the energy dependence of these neutral features is not a direct measure of the excited state dynamics. Instead, it is a measure of the slight differences in potential energies between the initial and final states in the core-to-valence transitions. From the simulation we find that the transient change in absorption energy observed within the first 40 fs of the experiment actually arises due to a potential well located in the core excited state potential energy surface.”* No change is made to the manuscript for this question.

4.- I assume that the potential energy curves depicted in Fig. 2c have been computed in the present work. If so, it should be mentioned in the figure caption and refer here to the supplementary material.

We have updated the caption of Fig. 3 to include a reference to the supplementary material where the calculation of the potential energy curves is described.

5.- Regarding process 2. The sentence in page 5 "Since the absorption peaks in Process 1 correspond to an excitation ... , as displayed in Fig. 4 (b).", appears a little unclear to me. Which is the relation with Process 1?

We thank the referee for pointing this out. There was a typo here and the sentence should read "Since the absorption peaks in Process 2 correspond to an excitation." **Thank you for pointing this out, we have edited it in the updated manuscript (pg. 5, paragraph 4).**

6.- A small typo in page 6 at the beginning of the second paragraph where I guess "moniter" should be replaced by "monitor".

Thank you for pointing out this typo, **we have edited it in the updated manuscript (pg. 6, paragraph 2).**

Response to Reviewer #2:

Title: Probing conical intersection dynamics with attosecond transient absorption spectroscopy

This manuscript describes what is presented as a attosecond transient absorption spectroscopic (ATAS) experiment in which the pump pulse is a strong-field 4fs NIR pulse, which prepares a complex wave packet (in the molecule CH₃Br) comprised of ground electronic but vibrationally excited states, electronically excited states, as well as multiple ionic continua. This wave packet is subsequently probed with a time-delayed XUV pulse that excites a 3d electron on the Br atom into the manifold of virtual orbitals (IP ~76 eV). The resulting time-evolution of the transient absorption spectrum is then analyzed to extract information about the initially prepared wave packet.

This work clearly shows the utility of using core spectroscopies to probe complex multi-state molecular dynamics. Each of the components of initially prepared wave packet are differentially projected out in the energy resolved spectrum. The NIR pump pulse will, in general, create a very complicated wave packet comprised of numerous electronic and vibrational components, which makes subsequent analysis of some time-resolved signal highly problematic. In this case, it precisely that characteristic of strong-field NIR excitation pulses that showcases the ability of the XUV pulse to disentangle and energy resolve all the different wave packet components.

We thank the reviewer for acknowledging the potential of attosecond core state spectroscopy and address the comments below.

However, there a couple issues with the manuscript as currently presented.

Firstly, the term ATAS is inappropriate as the dynamics they study is on the femtosecond time scale (as shown in their Figs). Their time resolution is also in the femtosecond, NOT sub-femtosecond (attosecond) time scale (as shown by their data). They should call it "ultrafast", not attosecond transient absorption.

The attosecond nature of the XUV probe pulse is quite important in the present experiment for three reasons: 1) It narrows the time resolution to below a single optical cycle of the NIR pump. If the XUV pulse was not a single attosecond pulse, then satellite pulses would appear at ± 1.3 fs, increasing the effective time resolution to > 2.6 fs. However, with the single attosecond pulse used here, we obtain an effective resolution below the NIR optical cycle. This enhances the resolution for both the non-adiabatic dynamics as well as the coherent spin-orbit wavepacket. 2) The broadband spectrum associated with a single attosecond pulse provides continuous coverage over a range of absorption features, providing similar SNR for each

absorption line. Without a single attosecond pulse, the periodic modulation of spectral intensity (due to harmonic contamination) gives rise to a varying SNR, which can effectively go to zero depending on how strong the harmonic modulation is. 3) Harmonic contamination can also give rise to a periodic background structure in the delta O.D. spectrum, limiting the visibility of weak broadband absorption signals. For these reasons, the attosecond nature of the XUV probe pulse is quite important.

Most “attosecond” experiments use a few-cycle NIR pulse and an attosecond XUV pulse and exhibit a cross-correlation width that is limited by the pulse duration of the few-cycle NIR field. While these “attosecond” experiments can probe a broad range of observables (i.e. attosecond time delays, sub-cycle oscillations, ensuing pump-probe signatures), the attosecond nature of the XUV pulse provides a unique tool for studying the shortest possible ultrafast science at play. For these reasons, the use of attosecond in referring to the type of spectroscopy in the manuscript is important, as the results would not be possible with usual ultrafast methodologies. **Therefore, no changes are made to the manuscript.**

Secondly, one notes that the experimental results are least compelling when it comes to the process that gives rise to the title of the manuscript: describing the passage through the conical intersection. The simulated spectrum clearly shows the signature of a bifurcation in the wave packet corresponding to an electronic transition, but the experimental spectrum does not. Rather, this passage is inferred from a delayed increase in a different dissociation channel -- although it's not obvious that a conical intersection is the sole explanation for that observation (i.e. dissociation resulting from direct preparation of the $1Q_1$ state). Furthermore, the supplementary material seems to suggest that this spectrum shown in Figure 3c was generated assuming that only $3Q_0+$ state was populated via the NIR pulse. This assumption makes the bifurcation of wave packet most clear in the figure, but is unlikely to rigorously hold. Confirming the validity of this assumption is possible via simulations of the NIR pump process, but they would be non-trivial.

The reviewer is correct in their assessment that the conical intersection is inferred due to the delayed increase of Br atomic yield and that we cannot unambiguously rule out direct dissociation from the $1Q_1$ state. **We have therefore included a sentence stating, “While contributions from the excitation and direct dissociation of the $1Q_1$ state cannot be unambiguously ruled out in the Br atomic yield, the remarkable agreement with the simulated dynamics and prior work strongly suggests that the observed wavepacket dynamics are directly associated with the presence of the conical intersection” (pg. 5, paragraph 1).** However, as discussed above, if we take a lineout of the transient absorption spectrum during the excitation of the neutral excited states (0.5 – 3.5 fs, plotted below), we observe a spin-orbit pair of absorption lines ($\Delta E_{SO} = 1$ eV, corresponding to the splitting of the core excited state) which we assign to $3Q_{0+} - D_{5/2}$ and $3Q_{0+} - D_{3/2}$ with a linewidth of ~450 meV (corresponding to the width of the excited state wavepacket). Our transient absorption simulation suggests that excitation to $1Q_1$ should appear as a pair of transitions ~0.4 eV below the $3Q_{0+}$ pair. However, no additional pair of absorption lines is observed in the experimental lineout. Therefore, within the limit of our experimental SNR, we do not observe another excited state in the initial excitation step. In addition, we know from previous experiments that the molecule is preferentially ionized along the perpendicular direction of the C-Br bond. At an

intensity of $1.5 \times 10^{14} \text{ W/cm}^2$, we are most likely depleting the available neutral population along this perpendicular axis. Therefore, the only states left to excite are the neutral parallel states. This strongly suggests that $^3Q_{0+}$ is the predominant neutral excited state in the present manuscript.

Further, the simulation of the transient absorption dynamics cannot handle a coherent superposition of excited states without explicit modeling of the light-matter interactions during excitation (which is computationally very demanding). We can incoherently add the dynamics of various states independently, however, this is clearly not the best way to perform the simulation. Therefore, by narrowing down the excited-state spectrum, we can more accurately reproduce the observed transient absorption dynamics.

Furthermore, They do not analyze the structures "during" the passage through the conical intersection. Their data show the 'before' and the 'after', not so different from the many other studies of internal conversion in molecules. The time scale may be faster here, but the result is the same: they see the 'before' and the 'after'.

This is a very good suggestion, unfortunately, our ability to directly observe bifurcation of the wavepacket through spectral splitting of the absorption feature is limited due to (i) the available spectral resolution ($\sim 200 \text{ meV}$) and (ii) the SNR of the measurement. While these parameters were optimized when the manuscript data was collected, an improved diffraction grating and a more stable CEP/spectral stability are necessary for ameliorating these issues.

In comparison, the assignments of the two other processes discussed are less ambiguous. The observation of the ground state vibrational wave packet and, in particular, the beats between the different ionic continua is relatively compelling. In general, the wealth of spectroscopic data amenable to assignment to the various components of the wave packet is impressive.

We thank the reviewer for acknowledging the quality of data for the vibrational and spin-orbit wavepacket.

In summary, the present work is first rate and merits publication in Nature Communications. However, the title of the manuscript focuses specifically on the one part of the experiment/analysis that is somewhat ambiguous. The nonadiabatic dynamics is inferred largely on the basis of numerical simulation and subsequent comparison to the experimental spectrum -- where former assumes that the prepared wave packet is simplified version of what is likely generated via the strong-field pump pulse.

That said, the attosecond XUV probe is shown to be a highly effective approach to simultaneously image multiple components of a complex molecular wave packet. These methods are leading edge and will be of general interest to the chemical physics community.

We have decided to change the title to, “Disentangling conical intersection and coherent molecular dynamics in methyl bromide with attosecond transient absorption spectroscopy,” to reflect the ability of attosecond core hole spectroscopy to resolve and disentangle the disparate molecular dynamics that the reviewer considers a highlight of the paper.

Specific Comments

1. In Figure 3b: the curve coloring implies adiabatic states, but the legend implies diabatic states. Also: the labels (b) and (c) should be switched.

We thank the reviewer for pointing out this error and **have correctly colored the diabatic states in Fig. 3 b in the updated manuscript.**

2. Some indication as to the nature/origin of simulated spectrum is warranted in the main body of the text.

The calculation of the simulated transient absorption spectrum is contained in the supplement and referenced to in the manuscript. Given the complexity of the calculation, any description would detract from the primary discussion in the manuscript. Therefore, we have chosen to keep the discussion of the transient absorption calculation within the supplement.

3. The discussion of the formation of a coherent wave packet comprised of ionic states is somewhat terse and difficult to understand. I suggest the authors more clearly state: i) what N-photon paths would be expected to interfere to generate the beat frequencies? ii) what level spacings would the present observable be sensitive to? iii) what specifically, is the origin of the single beat observed in the spectrum?

The interference that gives rise to the quantum beating does not arise from a multiphoton, quantum path interference (as would 2ω oscillations in many attosecond experiments). Instead, we are preparing a superposition of spin-orbit ionic states through the tunnel ionization bandwidth of the NIR pump pulse. The dynamics of the wavepacket are probed with core level transitions made by the XUV attosecond pulse and some of these transitions will land the two spin-orbit states in the same final state where they undergo interference with a period corresponding to the spin-orbit splitting of the two ionic states. We realize, such an explanation was missing in the manuscript, **therefore we have added a sentence explaining the modulation at the spin-orbit period as, “Further, provided that excitation of this coherent pair of spin-orbit ionic states arrives in the same final core-hole excited state upon XUV excitation (i.e. $3d_{3/2}^{-1}$), a modulation of the XUV transition probability will reveal the wavepacket dynamics”** (pg. 6, paragraph 1).

In addition, we realize the original description of the spin-orbit wavepacket as “rotating about the C-Br” axis is not exactly correct. Instead, it’s a modulation of the electron hole density between the p_x and p_y orbitals of the Br atom. **We have updated the description of the spin-orbit wavepacket in both the manuscript (pg. 6, paragraph 2) and the caption of Fig. 5.**

REVIEWERS' COMMENTS:

Reviewer #1 (Remarks to the Author):

In general terms, I am satisfied with the answers and changes included in the manuscript by the authors in this revised version considering my comments and suggestions and also those by the second reviewer. However, there are two points in which I must insist: First, at the end of the first sentence in page 2, the authors should add "...in real time for this molecule". Second, at the end of the last sentence of the third paragraph in page 5, the authors must change "...previously measured by..." by "...previously estimated by...". Once this changes are done I will recommend the paper to be publish without further review.

There is a typo in reference 9, it reads "& nares, L. B." and it should be "& L. Banares".

Reviewer #2 (Remarks to the Author):

The authors have sufficiently addressed the issues in the previous review and the manuscript is suitable for publication in Nature Communications.

REVIEWERS' COMMENTS:

Reviewer #1 (Remarks to the Author):

In general terms, I am satisfied with the answers and changes included in the manuscript by the authors in this revised version considering my comments and suggestions and also those by the second reviewer. However, there are two points in which I must insist: First, at the end of the first sentence in page 2, the authors should add "...in real time for this molecule". Second, at the end of the last sentence of the third paragraph in page 5, the authors must change "...previously measured by..." by "...previously estimated by...". Once these changes are done I will recommend the paper to be published without further review.

We have changed the last sentence on paragraph 1 of page 2 to read, "However, no study has directly probed these non-adiabatic dynamics in real time for CH₃Br." In addition we have updated the final sentence of paragraph 4 on page 5 to read "... previously estimated by Gougousi *et al.* [20] to be".

There is a typo in reference 9, it reads "& nares, L. B." and it should be "& L. Banares".

We have updated this typo in reference 9 and thank the reviewer for a thorough review of our manuscript.

Reviewer #2 (Remarks to the Author):

The authors have sufficiently addressed the issues in the previous review and the manuscript is suitable for publication in Nature Communications.